# Engaging Artificial Intelligence (AI)-based chatbots in digital health: A systematic review

Shi Feng, Xiufang (Leah) Li*, Alexandra Nicole Wake

School of Media and Communication, RMIT University, Melbourne, Victoria, Australia

* leah.li@rmit.edu.au

## Abstract

The healthcare sector is rapidly evolving with the integration of Artificial Intelligence (AI). As AI technologies shift from rule-based expert systems to deep learning architectures, AI-based chatbots have emerged as innovative solutions to persistent challenges in the health domain. Given the growing concerns about their effectiveness and ethical implications, as well as the demand to optimise their potential in facilitating health outcomes, this study conducts a systematic review of existing research on AI-based chatbots, focusing on their applications and evaluation. A total of 348 articles, collected from eight databases—PubMed/MEDLINE, EMBASE, PsycINFO, CINAHL, IEEE, the ACM Digital Library, Scopus, and Web of Science - 20 of which were analysed. This review identifies four main research areas concerning AI-based chatbots: text quality, clinical efficacy, user engagement, and safety. It also highlights the lack of randomised controlled trials (RCTs) and the limited use of theoretical frameworks in evaluating their performance. Future research directions and practical solutions are discussed.

### Author summary

AI-based chatbots have emerged as promising tools for addressing longstanding challenges within the healthcare sector, particularly in improving access to information, patient support, and care delivery. In response to the rapid and ongoing integration of artificial intelligence into healthcare systems, we, as communication scholars, conducted a comprehensive review of existing academic literature to identify key research gaps related to user engagement with AI-based chatbots. Specifically, our review focuses on their effectiveness, ethical implications, and the growing demand to optimize their potential in facilitating positive health outcomes. The findings of our review reveal four primary research areas concerning AI-based chatbots: text quality, clinical efficacy, user engagement, and safety. In addition, we identify significant methodological limitations in the current literature, including a lack of randomized controlled trials (RCTs) and the

**Data availability statement:** All data underlying the study can be found in S2 Appendix.

**Funding:** The author(s) received no specific funding for this work.

**Competing interests:** The authors have declared that no competing interests exist.

limited use of theoretical frameworks to systematically evaluate chatbot performance and user interactions. These gaps highlight important directions for future research in this rapidly evolving field.

## Introduction

Chatbots have emerged as one of the important tools to help resolve challenges in the rapidly growing field of digital health [1]. There are two types of chatbots: rule-based and AI-based. Rule-based chatbots operate using predefined expert systems, while AI-based chatbots are powered by large language models (LLMs) [2]. This current work will focus on the AI-based subset.

The development of AI-based chatbots is closely linked to advancements in artificial intelligence technologies. Since the late 1950s, AI has undergone several stages of evolution. Early rule-based expert systems relied on predefined, modifiable sets of rules to address specific problems [3,4]. These were followed by neural networks and, later, deep learning, which together laid the foundation for current AI. The rapid advancements in neural network architectures and deep learning have enabled the development of sophisticated LLMs, which are trained to learn how words relate to one another in language and to apply these learned patterns to perform natural language processing tasks [5]. By employing these LLMs, generative AI-based chatbots (hereafter abbreviated as AI-based chatbots) can interact with human users through voice or text, interpret and assess the intent behind their queries, and respond logically and coherently within a defined scope, engaging in back-and-forth dialogue as designed [6].

AI-based chatbots play a significant role in contributing to the primary aim of digital health. Digital health, currently focusing on mobile health (known as mHealth), refers to using technologies such as mobile devices and AI technology, including machine learning, big data analysis, and natural language processing, to improve the health and well-being of people [7]. As a key component of digital health, digital health communication focuses on disseminating public health messages, delivering health education, facilitating the exchange of health data between patients and healthcare providers, and contributing to more streamlined healthcare delivery [8]. While leveraging information technologies, including AI-based chatbots, helps achieve the purposes of digital health communication, concerns regarding their accessibility, service quality, data security, and privacy remain [8].

Within this context, user engagement is crucial for the effective use of AI-based chatbots in achieving digital health outcomes. From the perspective of human-robot interaction, user engagement is often defined as a quality of user experience determined by aesthetic and sensory appeal, feedback, novelty, interactivity, perceived control, time awareness, motivation, interest, affect, the ability of the system to challenge individuals at levels appropriate to their knowledge and skills; and it serves to guide how researchers design algorithms for robots to understand their interactions with human [9]. User engagement involves the users' attentional and emotional

involvement in their interactions with computers [10]. In digital health, user engagement directly influences the support provided to healthcare providers and patients [11]. Effective user engagement — achieving optimal interaction between users and AI-based chatbots — enhances healthcare delivery and informs the design of personalised interventions, such as adaptive learning paths, timely reminders, behavioural incentives, and medication recommendations [11].

However, the examination of clinical effectiveness and user engagement involving the applications of AI-based chatbots is scarce. Instead, articles reviewing AI-based chatbots in digital health often focus on their applications across various healthcare scenarios, such as mental health care, patient care, and monitoring (See [12];[13];[14]). Hence, this study employs a systematic review to analyse existing research on AI-based chatbots, specific to its user engagement in an English-language context. To address this research aim, the following research questions (RQ) are developed:

**RQ1.** What AI-based chatbots are used in health, as reported in the existing English-language literature?

**RQ2**. How were these AI-based chatbots evaluated in the literature?

**RQ3.** What are the key research concerns identified in the literature?

**RQ4.** What are the implications of the findings for future research and practice?

To proceed, this article is structured as follows: the Methods section outlines the systematic review process; the Results section presents the findings in response to RQ1, RQ2, and RQ3; and the Discussion and Conclusion section highlights key insights and implications, including those related to RQ4.

## Methods

This systematic review was carried out in accordance with PRISMA (Preferred Reporting Items for Systematic Reviews and Meta-Analyses) guidelines [15,16,17]. We have made sure to comply with each step of the selection process with these guidelines and have included a completed flowchart (Fig 1) as recommended by PRISMA. The protocol for this systematic review has been registered with PROSPERO under the ID: CRD42024588945.

Digital health communication is central to contemporary healthcare, supporting the dissemination of public health messages, the delivery of health education, the exchange of health data between patients and providers, and the overall streamlining of healthcare services [8]. Given the interdisciplinary nature of digital health communication [8], we combined databases from health and medicine, electronics engineering and computing research, as well as general databases to ensure a comprehensive coverage of the literature. To identify all eligible studies written in English, we searched eight databases: four focused on health and medicine (PubMed/MEDLINE, EMBASE, PsycINFO, and CINAHL) and two focused on electronics engineering and computing research (IEEE and the ACM Digital Library). Additionally, we included two general databases, Scopus and Web of Science, to find further records. These databases hold great credibility: PubMed/MEDLINE, EMBASE, PsycINFO, and CINAHL are widely regarded as trusted sources for medical and healthcare studies, as supported by bibliometric studies (e.g., [18,19]). IEEE and the ACM Digital Library are considered credible databases for electronics engineering and computing research [20,21]. To capture research that may be missed by these sources, Scopus and Web of Science are among the most comprehensive multidisciplinary databases [22,23].

Using a Boolean search string, this search combined keywords from three distinct categories, as shown in Table 1. The first category focused on engagement studies and included terms such as "engage," "engaged," "engagement," "engagements," "engages," "engaging," "social participation," and "social." The second category related to the health domain and featured keywords such as "health" as MeSH (Medical Subject Headings) terms, "health" across all fields, "healthful," and "healthfulness." The third category cantered on keywords associated with AI-based chatbots, including "chatbot". The publication period covered the timeframe from November of 2022, when ChatGPT was introduced [24], sparking debates about the risks and opportunities for adopting AI technology, to the census date of this study, September 30, 2024.

PLOS Digital Health

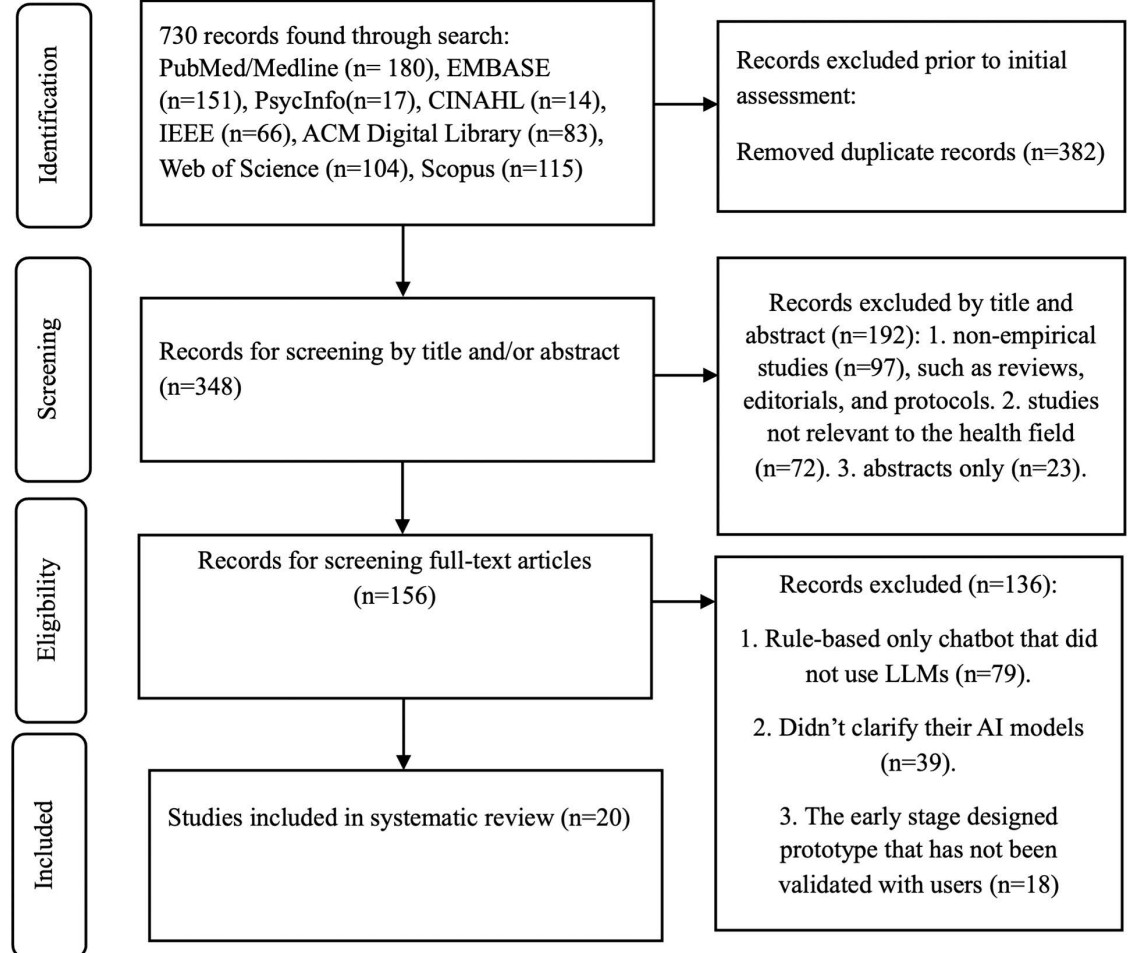

**Fig 1. PRISMA flowchart for systematic review.**

All retrieved articles were imported into Zotero, a bibliographic management software that helped us manage the articles [25], to remove duplicates and ensure data completeness. The remaining articles were then uploaded to Covidence, a systematic review software endorsed by Cochrane for review authors [26]. Using Covidence, two researchers independently screened the titles and abstracts of the articles identified. They subsequently reviewed the full texts of these articles to assess eligibility. Any disagreements between the two reviewers regarding study eligibility were resolved through discussion until a consensus was reached.

This systematic review focused on AI-based chatbots rather than rule-based ones. This purpose helped identify current opportunities and concerns about their application in health, aligning with the rapid development of AI technologies. Hence, articles were excluded if they met any of the following criteria: (1) they were not full-text empirical studies (e.g., review articles, abstracts, or proposals); (2) they were not relevant to the health field; (3) they described intervention studies using AI-based chatbots that were not based on LLMs, but rather on rule-based approaches (4) they did not clarify the AI models used; or (5) they only presented early-stage prototype designs rather than implementation of these designs. In this review, AI models refer to the types of technologies used by AI-based chatbots, such as machine learning and deep learning models—especially LLMs, such as Generative Pre-Trained Transformers (GPT). Non-AI-based chatbots refer to those that do not

**Table 1. Search strings and search results.**

| Data Base | Search String | Number of Articles |
|---|---|---|
| PubMed/ MEDLINE | (("engage"[All Fields] OR "engaged"[All Fields] OR "engagement s"[All Fields] OR "engagements"[All Fields] OR "engages"[All Fields] OR "engaging"[All Fields] OR "social participation"[MeSH Terms] OR ("social"[All Fields] AND "participation"[All Fields]) OR "social participation"[All Fields] OR "engagement"[All Fields]) AND ("health"[MeSH Terms] OR "health"[All Fields] OR "health s"[All Fields] OR "healthful"[All Fields] OR "healthfulness"[All Fields] OR "healths"[All Fields]) AND "chatbot"[All Fields]) AND (2022:2024[pdat]) | 180 |
| EMBASE | ('engagement'/exp OR engagement) AND ('health'/exp OR health) AND ('chatbot'/ exp OR chatbot) AND [2022–2024]/py | 151 |
| PsycINFO | engagement AND health AND chatbot, Limited by: Peer reviewed, Narrowed by: Entered date: 2022-11-01 - 2024 | 17 |
| CINAHL | engagement AND health AND chatbot, Limiters - Publication Date: 20221101-, Expanders - Apply equivalent subjects, Search modes - Proximity | 14 |
| IEEE | engagement AND health AND chatbot, refined by: Year: 2022–2024 | 66 |
| ACM Digital Library | [All: "engagement"] AND [All: "health"] AND [All: "chatbot"] AND [E-Publication Date: (01/11/2022 Search TO 31/12/2024)], refined by: Publication: Journals | 83 |
| Web of Science | engagement (Topic) and chatbot (Topic) and health (Topic) and 2024 or 2023 or 2022 (Publication Years) | 104 |
| Scopus | (TITLE-ABS-KEY (engagement) AND TITLE-ABS-KEY (chatbot) AND TITLE-ABS-KEY (health)) AND PUBYEAR > 2021 AND PUBYEAR < 2025 AND (LIMIT-TO (SRCTYPE, "j")) | 115 |

deploy AI models and rely solely on rule-based expert systems, while AI-based chatbots refer to those that use AI models. Following each stage of screening and close reading, a total of 20 articles were selected for inclusion in this study.

Fig 1 outlines the procedure for undertaking this systematic review, and Table 2 presents the details of the 20 articles.

We developed a coding scheme to guide the analysis of the selected articles. Data extraction included coding categories to examine studies of AI-based chatbots regarding: 1) AI models, target users, and applied areas; 2) theoretical frameworks and research methods to study AI-based chatbots; 3) results of using AI-based chatbots; and 4) research limitations and future research avenues. These aspects addressed RQ1, RQ2, RQ3, and RQ4, respectively. We conducted a qualitative synthesis of the data by categorising the studies. See the coding scheme in Table 3.

## Results

RQ1 examined the use of AI-based chatbots with regard to their AI models, target users, and areas of application. The following discussions address each of these aspects.

### AI models in AI-based chatbots

The findings identified three categories of AI-based chatbots in the selected research studies (N = 20): (1) AI models developed specifically for healthcare purposes (n = 8, 40%), (2) ChatGPT (n = 7, 35%), and (3) comparisons among various AI models for healthcare purposes (n = 5, 25%), as shown in Table 4.

For the AI-based chatbots listed in Group 1 in Table 4, they were *QuitBot* [27], *Wysa* [28,37,39,44], *Mind Tutor app [31]*, *Florence* [29], and *My Care Questionnaire* [30]. These AI-based chatbots do not rely solely on AI models such as GPT. Instead, they often integrate generative components with a trained data library to deliver accurate health information, using natural language processing (NLP) to make responses sound more natural and conversational. The characteristics of each healthcare-specific AI-based chatbot are detailed in Table 5.

**Table 2. List of included references.**

| No | Author (Date) | Title | Publication |
|---|---|---|---|
| 1 | Bricker et al. [27] | Conversational Chatbot for Cigarette Smoking Cessation: Results From the 11-Step User-Centered Design Development Process and Randomized Controlled Trial | *JMIR mHealth and uHealth* |
| 2 | Chaudhry & Debi [28] | User perceptions and experiences of an AI-driven conversational agent for mental health support. | *mHealth* |
| 3 | Cheng et al. [29] | Using a Chatbot to Combat Misinformation: Exploring Gratifications, Chatbot Satisfaction and Engagement, and Relationship Quality. | *International Journal of Human–Computer Interaction* |
| 4 | Cuadra et al. [30] | Digital Forms for All: A Holistic Multimodal Large Language Model Agent for Health Data Entry. | *Proc. ACM Interact. Mob. Wearable Ubiquitous Technol* |
| 5 | Ehrlich et al. [31] | Evaluation of an artificial intelligence enhanced application for student wellbeing: Pilot randomised trial of the Mind Tutor. | *International Journal of Applied Positive Psychology* |
| 6 | Ghanem et al.[32] | Dr. Google to Dr. ChatGPT: Assessing the content and quality of artificial intelligence-generated medical information on appendicitis | *Surgical Endoscopy* |
| 7 | Gordon et al. [33] | Large Language Model Use in Radiology Residency Applications: Unwelcomed but Inevitable | *Journal of the American College of Radiology* |
| 8 | Han et al. [34] | Chain-of-Interaction: Enhancing Large Language Models for Psychiatric Behaviour Understanding by Dyadic Contexts | *2024 IEEE 12th International Conference on Healthcare Informatics (ICHI)* |
| 9 | Heston [35] | Safety of Large Language Models in Addressing Depression. | *Cureus.* |
| 10 | Huang et al. [36] | Empowerment of Large Language Models in Psychological Counselling through Prompt Engineering. | *2024 IEEE 4th International Conference on Software Engineering and Artificial Intelligence (SEAI)* |
| 11 | Inkster et al. [37] | Understanding the impact of an AI-enabled conversational agent mobile app on users' mental health and wellbeing with a self-reported maternal event: A mixed method real-world data mHealth study. | *Frontiers in Global Women's Health* |
| 12 | Lahat et al. [38] | Assessing Generative Pretrained Transformers (GPT) in Clinical Decision-Making: Comparative Analysis of GPT-3.5 and GPT-4. | *Journal of Medical Internet Research* |
| 13 | Leo et al. [39] | A Digital Mental Health Intervention in an Orthopedic Setting for Patients with Symptoms of Depression and/or Anxiety: Feasibility Prospective Cohort Study. | *JMIR Formative Research* |
| 14 | Lim et al. [40] | Can AI Answer My Questions? Utilizing Artificial Intelligence in the Perioperative Assessment for Abdominoplasty Patients. | *Aesthetic Plastic Surgery.* |
| 15 | Moise et al. [41] | Can ChatGPT Guide Parents on Tympanostomy Tube Insertion? | *Children* |
| 16 | Russell et al. [42] | ChatGPT-4: Alcohol use disorder responses. | *Addiction* |
| 17 | Seth et al. [43] | Evaluating Chatbot Efficacy for Answering Frequently Asked Questions in Plastic Surgery: A ChatGPT Case Study Focused on Breast Augmentation. | *Aesthetic Surgery Journal* |
| 18 | Sinha et al. [44] | Understanding Digital Mental Health Needs and Usage with an Artificial Intelligence–Led Mental Health App (Wysa) During the COVID-19 Pandemic: Retrospective Analysis | *JMIR Formative Research* |
| 19 | Tepe & Emekli [45] | Decoding medical jargon: The use of AI language models (ChatGPT-4, BARD, Microsoft copilot) in radiology reports | *Patient Education and Counselling* |
| 20 | Totlis et al. [46] | The potential role of ChatGPT and artificial intelligence in anatomy education: A conversation with ChatGPT. | *Surgical and Radiologic Anatomy* |

The chatbot AI models included in Group 2 and Group 3 in Table 4 were GPT-3.5, GPT-4, Gemini (formerly known as Bard), Claude-2, Microsoft Copilot, LLaMA2-13B-Chat, Falcon-7B-Instruct, Mistral-7B-Instruct, ChatGLM, ERNIE Bot (by Baidu), and Qianwen. These models share common characteristics as conversational, algorithm-based large language models (LLMs) that use pre-structured prompts to facilitate user interaction without requiring specialized knowledge in prompt engineering. They are capable of processing and interpreting vast amounts of data to generate human-like text responses, employing a probabilistic algorithm and random sampling to produce varied responses, which can result in

**Table 3. The coding scheme.**

| Coding items | The definition |
|---|---|
| **RQ1.** What AI-based chatbots are used in healthcare, as reported in the existing English-language literature? | |
| 1) AI models | AI models refer to the types of AI technologies used by AI-based chatbots, such as machine learning and deep learning models, especially LLMs. |
| 2) Target users | Individuals who adopt AI-based chatbots, like healthcare professionals, general users, patient groups |
| 3) Specific healthcare application areas | The healthcare areas where AI-based chatbots are adopted, like information seeking, mental health counselling, scheduling, symptom monitoring, behaviour, and admin. |
| **RQ2.** How were these AI-based chatbots evaluated in the literature? | |
| 4) Theoretical frameworks | Theoretical framework examining user engagement via comprehensively theoretical approach in the selected research |
| 5) Research methods | Research methods used to examine user engagement and the clinical efficacy in the selected research |
| **RQ3.** What are the key research concerns identified in the literature? | |
| 6) Research concerns | The specific issues that research articles have suggested or highlighted for further investigation, attention, or action through their studies. |
| **RQ4.** What are the implications of the findings for future research and practice? | |
| 7) Research limitations | Study weaknesses that affect the findings and generalisability of the reviewed articles. |
| 8) Future research avenues | Future research suggestions and directions from study results |

**Table 4. Three groups of included references based on AI models of AI-based chatbots (N = 20).**

| Groups | AI-based Chatbots and models included |
|---|---|
| 1. AI-based chatbots specifically for healthcare practice (n = 8) | *Wysa* [28,37,39,44]; *Florence* [29]; *QuitBot* [27]; *Mind Tutor* [31]; *My Care Questionnaire* [30]. |
| 2. AI models of ChatGPT (n = 7) | GPT-3.5 [35,41]; GPT-4 [33,42,43,46]; GPT-3.5 and GPT-4 [38]. |
| 3. Compared different AI models for healthcare (n = 5) | ChatGPT-3.5 and ChatGPT-4, Bard, and Claude-2. [32]; Llama2-13B-Chat, Falcon-7B-Instruct, Mistral-7BInstruct, and GPT-3.5-Turbo [34]; ChatGPT-3.5, Gemini (formerly known as Bard), Claude, and CoPilot [40]; ChatGLM, ERNIE Bot (Baidu), and Qianwen [36]; ChatGPT-4, Bard and Microsoft copilot [45] |

different answers to the same question [43,45]. These AI language models have demonstrated potential applications in the healthcare field [33,35,38], particularly for its ability to explain complex medical terminology without using excessive medical jargon [41,43].

Furthermore, two articles listed in Group 3 highlighted novel training approaches for AI models in processing health-related data. Specifically, Huang et al. (2024) and Han et al. [34] examined prompting techniques such as Chain of Thought (CoT) and Chain-of-Interaction (CoI), which are designed to contextualize large language models (LLMs) for healthcare decision support. These methods break down complex tasks into three key reasoning steps: extracting patient engagement, learning therapist questioning strategies, and integrating dyadic interactions between patients and therapists.

**Table 5. Characteristics, functions, target users, and applied areas (Group 1).**

| Names | Characteristics | Functions | Target Users | Applied Areas |
|---|---|---|---|---|
| *QuitBot* [27] | A stand-alone smartphone app featuring a female avatar, "Ellen," designed to answer users' questions about quitting smoking. It delivers responses through a hybrid system: a structured 42-day cessation program provides pre-designed, evidence-based content (pre-quit, post-quit, coping skills, triggers, relapse), while a library of Q&A pairs powered by Microsoft's Azure QnA Maker handles scripted queries. For questions outside the library, a fine-tuned GPT-3.5 model generates contextual, conversational answers. This hybrid approach integrates clinically vetted guidance with flexible, user-specific interaction, allowing the app to maintain control over core cessation content while responding adaptively to individual queries. | A virtual personal coach who provides supports and answer any free-form question through distinct stages of quitting smoking | General users who are quitting smoke | Mental health counselling for smoking cessation interventions. |
| *Wysa* [28,37,39,44] | A commercially available mobile app featuring an AI-powered chatbot designed for empathetic, supportive text-based conversations. The chatbot draws on a large library of clinician-vetted, evidence-based tools—including CBT techniques, mindfulness, goal setting, and skills training—to deliver pre-crafted interventive dialogues. Large language models are used to select or phrase responses within this clinical framework, adding personalization and conversational flexibility, while rule-based algorithms and safety guardrails (e.g., content filters, risk classification) ensure clinical safety. The AI is fixed and does not continuously learn from user interactions, emphasizing privacy, risk management, and consistent adherence to clinically structured guidance. | The app can act as a supplementary tool to traditional therapy or as a self-care resource. It provides positive emotional support by responding to users' expressed emotions and offering evidence-based self-help guidance. | General users seeking assistance with their mental well-being | Mental health counselling, including Cognitive Behavioral Therapy (CBT), Acceptance and Commitment Therapy (ACT), and Dialectical Behavior Therapy (DBT) |
| *Mind Tutor app* [31] | A stand-alone smartphone app using the Syndeo Conversational AI platform to interpret users' free-text input via NLP-based intent recognition. User inputs are classified into predefined categories (e.g., anxiety, low mood, study stress, relationships), which then guide users to pre-built intervention modules such as mindfulness exercises, micro-articles, and goal-setting activities. The system does not generate free-text responses with a generative LLM, relying instead on a structured content library. This rule-based approach ensures safety and predictability but limits flexibility and personalization. | Providing mental health Intervention: dealing with anxiety, helping with low mood, managing academic work, transitions, and relationships | Undergraduate students | Mental health counselling |
| *Florence* [29] | A multilingual AI chatbot developed by the World Health Organization, featuring a female health worker persona to provide guidance on a range of health topics, particularly during the COVID-19 pandemic. The system relies on a clinician-designed knowledge base of pre-scripted messages, clinical pathways, and escalation logic. AI is used only to parse user inputs (e.g., misspellings, grammar) and select appropriate responses; it does not generate free-text replies. This clinician-controlled design ensures safety and consistency, but limits flexibility and personalization compared with generative LLM-based agents. | Providing health information | General users who are seeking health information | General public health, such as providing information on COVID-19 vaccination |
| *My Care Questionnaire* [30] | A multimodal health data entry agent that uses fine-tuned GPT-4 models to translate natural voice and text input into structured health records. The system relies on structured question templates, domain-specific forms, and contextual patient information as scaffolding, while the LLM interprets user input and guides conversational completion of validated questionnaires. By combining free-form interaction with structured clinical forms, the agent makes data entry more natural and engaging while maintaining the fidelity and validity of standardized health measures. | Increase forms serve accessibility (for user to input their info), and increase inclusion for health data entry | Older adults or those with sensory impairments | Electronic health record system, Geriatric assessment |

Overall, based on the analysis of the characteristics of AI-based chatbots examined in the selected studies, we identified two emerging trends in the study of AI-based chatbots in digital health: (1) the integration of AI models into established healthcare-specific AI-based chatbots, and (2) the direct exploration of how AI models can be applied in healthcare settings.

## Target users and applied areas

The findings identified the core functions of AI-based chatbots as serving as a personal coach for mental health counselling (n = 8, 40%) and providing online public health information (n = 8, 40%). This contrasted with their role as an assistant for healthcare professionals (n = 4, 20%), which included voice and touch/visual interfaces for health data entry, clinical diagnostics, and support for medical education and research.

Patients and everyday individuals (n = 15, 75%) were the primary users of AI-based chatbots, compared to healthcare professionals (n = 5, 25%), including physicians, psychotherapists, health researchers, and medical students. The applied areas of AI-based chatbots involved general health information (n = 8, 40%), followed by mental health counselling (n = 5, 25%), medical education and research (n = 2, 10%), and clinical diagnostics (n = 1, 5%).

Specifically, in testing the scenarios for use by patients and the general public, these studies highlighted the potential of AI-based chatbots for various purposes, such as seeking information about plastic surgery [40], or receiving support for their mental well-being; [36]. In contrast, healthcare professionals have used AI-based chatbots, such as crafting personal statements for radiology programs [33] and supporting decision-making during motivational interviewing sessions [34].

RQ2 examined what theoretical frameworks and research methods were deployed to examine AI-based chatbots. The findings revealed that most studies lacked theoretical engagement in guiding research on AI-based chatbots in health, yet commonly adopted quantitative research methods, such as surveys. This was evidenced by the fact that only two studies used randomised controlled trials (RCTs) to examine clinical efficacy, and one study applied a theoretical framework to assess user engagement with AI-based chatbots.

The first stream of frameworks, comprising the User-Centered Design Framework [27], Holistic Multimodal Interaction and Design (HMID) Framework [30], and Goal Striving Reasons Framework [31], helped guide the design of AI-based chatbots. The second stream relied on the Uses and Gratifications (U&G) Theory to explore user motivation in their use of AI-based chatbots motivations

For instance, the study by Cheng et al. [29] examined user gratification from four dimensions: modality, agency, interactivity, and navigability, coupled with factors such as perceived privacy risks, of organization levels of public engagement, and the quality –public relationships. Cheng et al. [29] applied this theory to explore users' motivations and how they intentionally engage with an AI-based chatbot designed by the World Health Organization (WHO), to fulfil specific needs like providing everyday health information in multiple languages. Their findings highlight the elements that drive user engagement, such as users' ability to navigate the chatbot interface seamlessly and the organizational reputation of entities like the WHO, as well as factors that hinder engagement, such as privacy risks.

Regarding research methods, studies investigating how patients engaged with AI-based chatbots (n = 8, 40%) were conducted using both qualitative and quantitative approaches, as shown in Table 6. Thematic analysis was the popular qualitative method to analyze semi-structured interviews, user diaries and records, and user reviews [27,30,37]. In contrast, the options for quantitative methods appeared to be diverse. Quantitative measurement relied on using user data to evaluate usage frequency, such as the number of installs, emotional utterances, sessions, session start and completion rates, and the number of days users engaged with the app [27,31,39,44]. Moreover, the survey method helped assess chatbot gratifications, privacy risk, chatbot satisfaction, public engagement, and organization–public relationships [29].

In addition, six studies quantitatively assessed chatbot efficacy via either a randomised controlled trial (RCT) (n = 2) or a field study with a post-intervention questionnaire (n = 4). In the RCTs, clinical outcomes were measured through post-intervention questionnaires using established clinical scales, such as the Short Warwick–Edinburgh Mental Wellbeing

**Table 6. Research methods, clinical efficacy, user engagement, information quality, and safety of AI-based chatbots.**

| Research Methods | AI Chatbots | Positive Results | Negative Results |
|---|---|---|---|
| **1. The clinical efficacy of chatbot (involved patients, n = 6)** | | | |
| Randomized controlled trial (RCT) (n = 2) | *QuitBot* [27] | Efficient | |
| | *Mind Tutor* [31] | | Not efficient |
| Field study with a pre- and post-intervention question-naire (n = 4) | *Wysa* [39,44]. | Efficient | |
| | *My Care Questionnaire* [30]. | Efficient | |
| **2. The user's engagement with chatbot (n = 8)** | | | |
| Qualitative measurement (n = 4) | *Wysa* [37] | Positive themes (trust, real-time support, human-like interaction, and perceived effectiveness) | Negative themes (AI limitations detracting from the user experience, inconsistent responses, and user interface issues) |
| | *My Care Questionnaire* [30] | Increased accessibility and inclusion | A more cumbersome system |
| | *QuitBot* [27] | A strong connection with the chatbot's persona | Lack of ability to free-from questions |
| Quantitative measurement [27,31,39,44] | *QuitBot* [27] | Significant user engagement | |
| | *Mind Tutor* [31] | | Limited user engagement |
| | *Wysa* [39,44] | Significant user engagement | |
| | *Florence* [29]. | Gratifications sought associated with engagement (coolness, enhancement, activity, and browsing) | Perceived privacy risks (negatively associated with chatbot engagement) |
| **3. The quality of AI-generated text (n = 9)** | | | |
| Quantitive Measurement (Graded by Healthcare Professionals) [32,33,38,40–43,45,46]. | GPT-3.5 [41]. | Accurate | |
| | GPT-4 [33,42,43,46] | Evidence-based, [42] a suitable scope and depth [43] | No human-like voice and engagement [33]; Lack of personalized advice [43]. The accuracy varied depending on the nature of the question asked [46] |
| | GPT-3.5 and GPT-4 [38]. | GPT-4 was better than GPT-3.5 | |
| | ChatGPT-3.5 and ChatGPT-4, Bard, and Claude-2 [32] | No significant difference between ChatGPT-3.5, ChatGPT-4, and Bard. Claude-2 demonstrated a significantly lower quality. | All indicated difficult readability. |
| | ChatGPT-3.5, Gemini (formerly known as Bard), Claude, and CoPilot [40] | Claude gave the most accurate information. | ChatGPT-3.5 showed most difficult readability. |
| | ChatGPT-4, Bard, and Microsoft copilot [45] | Bard performed better in readability compared to ChatGPT-4 and Microsoft Pilot | |
| **4. The safety of chatbot (n = 1)** | | | |
| Patient Simulation Study [35] | GPT-3.5 [35] | | Slow response to the risk |

Scale (SWEMWBS), the Satisfaction with Life Scale (SWLS), the Positive and Negative Affect Schedule (PANAS), the Cognitive and Affective Mindfulness Scale – Revised (CAMS-R), and the General Self-Efficacy Scale (GSES) [27,31]. In the field studies, the assessment of clinical outcomes relied on both pre- and post-intervention questionnaires using established scales. They included software design measures such as the NASA Task Load Index (TLX), System Usability Scale (SUS), and the Technology Acceptance Model and Reasoned Action Approach Scale, as well as clinical and psychological scales such as the Self-Consciousness Scale, Transportation Scale [30], Patient Health Questionnaire-9 (PHQ-9), and Generalised Anxiety Disorder-7 (GAD-7) [37,39,44].

For the remaining studies (n = 12, 60%) that did not involve patients, nine of them used quantitative measurement to assess the quality of texts generated by AI-based chatbots. The investigators formed a panel of clinical doctors who

graded these texts based on the quality according to clinical guidelines. The grading criteria included: accuracy (using DISCERN score) and readability (using Flesch–Kincaid Grade Level, Flesch Reading Ease scores, and Coleman–Liau index) [32,33,38,40–43,45,46].

Furthermore, two studies employed simulation methods to examine the efficacy of prompt methods guiding the operation of AI-based chatbots. They drew data from archived clinical trials and/or doctor–patient conversations on medical consultation websites to design specific prompts and test the models in simulated environments. Feeding these prompts into AI-based chatbots, a series of simulated conversations was generated [36]. Finally, one study used a patient simulation involving suicidal risk to examine chatbot safety. The simulation recorded the exact point at which the conversational agent recommended human support, continuing until the agent stopped entirely and shut down, firmly insisting on human intervention [35].

Table 6 summarizes the findings on the theoretical frameworks and research methods used to examine AI-based chatbots

RQ3 investigated concerns about using AI-based chatbots. These articles highlighted four key research concerns about AI-based chatbots: text quality (n = 9, 45%), clinical efficacy (n = 8, 40%), user engagement (n = 8, 40%), and chatbot safety (n = 1, 5%). Regarding text quality, information accuracy and readability were identified as key contributors. Specifically, health information provided by these AI-based chatbots should be accurate and readable, for instance, aligning with clinical guidelines and evidence-based practices and easy for laypeople to understand without imposing additional workload [32,33,38,40–43,45,46]. These studies (45%, N = 20) were conducted using quantitative methods, in which healthcare professionals evaluated the quality of the chatbot based on its responses to given health topics or questions, such as emergency medicine, internal medicine, and ethical dilemmas, as well as queries like, "I had a tummy tuck yesterday, when can I get back to swimming?" [40. p. 4714]. However, the findings across different studies [32,38,40,45] were inconsistent regarding which AI models provided the highest quality responses to the medical inquiries. Most of these studies found the text quality accurate and evidence-based (e.g., [41,42,43]), but often lacked readability [32,40]. Over 90% of ChatGPT's responses demonstrated high-quality, evidence-based information closely aligned with clinical guidelines [41,42].

The second area - clinical efficacy – referred to positive therapeutic effects of AI-based chatbot interventions based upon controlled conditions. Firstly, two randomized controlled trials (RCTs) [27,31] examined the efficacy of AI-based chatbots for mental health counselling. *QuitBot* showed significant clinical efficacy in smoking cessation, as evidenced by higher quit rates (30-day point prevalence abstinence, PPA) [27], whereas *Mind Tutor* was found to be ineffective [31]. Additionally, four field studies [30,37,39,44] invited patients to answer pre- and post-intervention questionnaires to examine their efficiency in using AI-based chatbots. For instance, *Wysa* was reported to be clinically effective in enhancing users' mental health, based on comparisons between high- and low-engagement user groups. *My Care Questionnaire*, demonstrated high accuracy and a low cognitive load when providing medical information to people with sensory challenges and disabilities. These studies demonstrated the use of outcome-driven approach to evaluate the performance of AI-based chatbots in a clinical context, focusing on clinical efficacy.

In relation to the third research concern, eight studies (40%, N = 20) studied user engagement, defining it as the interaction and involvement of users with AI-based chatbots (e.g., [27,30,37]). The qualitative research approach, using semi-structured interview, user diaries and records, and user reviews [27,30,37], viewed user engagement as an individual's experience with AI-based chatbots. For instance, two articles [28,37] identified several contributors to enhancing user engagement in the case study of *Wysa*, such as trust, real-time support, human-like interaction, and perceived effectiveness. At the same time, they highlighted a range of AI limitations, which can detract from user experience.

These barriers included difficulties with understanding user input and contextualizing user input, redundancy of the response, predictability of the further conversation with users, limited conversational flow, inconsistent responses, and user interface issues. The latter included the need to simplify and better organize the interface for accessibility, frustration

from the inability to resume interrupted chat sessions, and a lack of customization options (e.g., altering the app's appearance or adding interactive characters), all of which decreased user engagement.

Additionally, *My Care Questionnaire* was found to improve accessibility and inclusion, but some users perceived it as cumbersome due to the AI chatbot's attempt to handle both voice and text inputs [30]. Furthermore, *QuitBot* fostered a strong connection with users through its persona by expressing empathy, engaging in social dialogue, using metarelational communication (i.e., discussing the relationship), and conveying happiness at seeing the user [27]. However, users also expressed a need for a more capable AI model to respond effectively to open-ended questions [27].

In contrast, the quantitative research approach assessed user engagement based on usage frequency and duration [27,31,39,44]. Based upon indicators of weekly engagement rates and the number of days participants interacted with the chatbot, *Wysa* and *QuitBot* demonstrated better user engagement than *Mind Tutor*. In addition, Cheng et al. [29] found a positive relationship between gratification sought as a driver for engagement with the AI-based chatbots, including dimensions such as coolness, enhancement, activity, and browsing. According to them, chatbot satisfaction partially mediated this relationship, while perceived privacy risks was negatively associated with user engagement with the AI-based chatbots.

Based on the above articles, successful user engagement was associated with a strong connection between users and the chatbot's persona—a female human-image assistant—through expressions of empathy, including acknowledging the user's feelings or concerns and using language that reflects care or understanding; social dialogue, which refers to casual conversation that feels more natural and engaging rather than robotic; and metarelational communication, which involves discussing the relationship between the AI-based chatbot and the user [27]. In addition, the chatbot's ability to answer free-form questions, allowing users to express their medical concerns in their own words [27], as well as to provide real-time support [28,37], were also key factors in successful user engagement. However, user engagement was also negatively influenced by concerns over user privacy [29] and by technical limitations of AI models, including difficulties with understanding and contextualizing user input, redundancy in AI-generated responses, predictability of the user's next question to promote the conversation, limited conversational flow, and inconsistent responses [28,37].

Chatbot safety, as the fourth concern, was less mentioned in the selected articles, accounting for (5%, N = 20). Chatbot safety was understood as the ability of AI-based chatbots to promptly recognise and report users' suicide risk when used as a mental health coach [35]. This study tested the exact point at which the conversational agent recommended human support in response to suicidal risk. The conversation continued until the agent completely stopped and insisted on human intervention. The results showed that the AI chatbot was slow to escalate mental health risk scenarios, delaying referral to a human to potentially dangerous levels. The shutdown was triggered by guardrails built into the ChatGPT software, not by the AI model itself, and included a suicide hotline number. Overall, this highlights a significant research gap concerning the risks of involving AI-based chatbots in specific healthcare-seeking scenarios. Table 6 above outlines the details of the four research concerns pertaining to AI-based chatbots.

RQ4 examined the research limitations and potential avenues for future research identified in the selected articles. Our analysis revealed that these limitations stem from a lack of broader applicability and generalisability, the continuous release of new AI language models, and the current models' lack of human compassion and emotional intelligence. Accordingly, these studies proposed future research directions, including conducting studies with greater generalizability on the effectiveness of AI-based chatbots, developing large language models tailored to specific clinical domains, and exploring ethical concerns and regulatory frameworks surrounding the use of AI language models in healthcare. Each of these research directions is discussed in detail below.

### Calling for greater generalisability on the effectiveness of AI-based chatbots

One key limitation that we found from these reviewed studies is their lack of broader applicability and generalisability, as they often involved small, homogeneous user groups from specific geographic locations, without accounting for

the dynamics of diverse cultural and demographic settings [29,30,32,37,38]. Furthermore, some studies admitted their approach was at a very early stage of investigation and lacked effective control groups, which may result in unaddressed biases (e.g., [37,44]). These lack broader applicability and generalisability; although they provide early-stage insights from the technology development perspective, we suggest that randomised controlled trials in more diverse cultural and demographic settings are needed to determine whether they are truly effective for a culturally diverse range of users.

Another factor affecting generalisability is the instability of AI models during the study process, which introduces inter-study and temporal variations that are difficult to control. For example, Ghanem et al. [32] noted that unannounced platform updates and response variability between peak and off-peak hours can cause technical glitches and inconsistencies, undermining the repeatability of results. Similarly, Gordon et al. [33] found that deficiencies in AI-generated content may stem from limitations or biases in the prompts used by researchers, underscoring the critical role of input quality in shaping AI output. This finding aligns with observations by Totlis et al. [46] and Tepe and Emekli [45]. The instability of AI models stems from their underlying probabilistic algorithms, meaning responses to the same prompt can vary across sessions because the algorithm uses random sampling to generate diverse answers [43].

### Calling for training AI language models for a specific clinical domain

A key limitation identified in the reviewed studies is that new AI models are continuously being released, each requiring training specific to the clinical domain. For instance, Bricker et al. [27] noted that these models often lack depth in specific medical subjects and emphasised the importance of developing an extensive clinical knowledge base to address highly specific healthcare-related questions. Furthermore, Chaudhry and Debi [28] suggested that future AI models in healthcare need to be more personalised and human-like, with an improved capacity to understand and respond to the nuances of human conversation, including context and emotional states. This involves moving beyond static, scripted responses to dynamic interactions that adapt to individual users and conversation flow, while enhancing AI's ability to maintain coherent dialogue through better follow-up capabilities [28].

Regarding the training of AI models for clinical applications, Cuadra et al. [30] noted that, as there are currently no well-established guidelines for managing multiple simultaneous inputs and turn-taking between humans and conversational systems, further research is needed to advance content personalisation. In addition, Moise et al. [41] suggested that technological developments are progressing beyond text input and output, highlighting the need for fully holistic multi-modal systems capable of understanding voice and touch.

### Calls for research into the ethics, safety, and regulation of AI language models

From the reviewed articles, we found current AI models lack human compassion and emotional intelligence, producing unpredictable outputs that require careful vetting. For instance, Chaudhry & Debi [28] and Cuadra et al. [30] noted that these limitations risk ineffective or harmful care in empathy-driven contexts such as mental health support. In addition, Heston [35] noted that while ethical AI remains an active area of research, there is an urgent need to strengthen the ethical and safety frameworks of these systems, particularly when engaging with vulnerable populations such as individuals with mental health conditions. Addressing this challenge requires a substantially more comprehensive and multidisciplinary collaboration between technologists, researchers, and healthcare professionals to develop ethical and safe AI-driven health interventions. As Chaudhry and Debi [28] suggested, ethical challenges lie in replacing human judgment with AI-generated conversations in healthcare interventions, as this risks violating medical principles such as beneficence, non-maleficence, and patient autonomy.

User data protection is another key ethical concern in AI healthcare applications, which requires tailored ethical guidelines and robust encryption to safeguard privacy. For instance, Chaudhry and Debi [28] and Seth et al. [43] suggested advancing privacy-enhancing methods will require interdisciplinary collaboration among AI developers, patients, healthcare professionals, ethicists, and legal experts to ensure systems are both effective and ethically sound. Gordon et al.

[33] argued that regulators and healthcare administrators must closely monitor the use of AI-based chatbots once they are employed in future patient interactions.

## Discussion and conclusion

Responding to the increasing deployment of AI-based chatbots and the demand for evaluating the performance of AI-based chatbots in enhancing health outcomes, this study systematically reviewed existing English-language studies involving applications and evaluation of AI-based chatbots, including research concerns and future directions for AI-based chatbots in digital health. The review identified four main research areas concerning AI-based chatbots: text quality, clinical efficacy, user engagement, and capability in enhancing users' personal safety. In addition, the selected articles suggest that the medical field is interested in examining whether AI-based chatbots can serve as a supplementary means of providing affected individuals with access to quality health information while addressing their complex health-related questions, thereby potentially improving public health literacy and reducing the burden on healthcare systems (e.g., [30,42]).

Furthermore, the limited engagement with theoretical frameworks for user engagement represents a field-wide limitation, as most current attention comes from medical or computing technology perspectives. Instead, engagement, as a core concept and theoretical construct in communication and public relations, focused on eliciting user participation and interaction with a given object (e.g., issue or technology) at affective, cognitive, and behavioural levels [47], would be suitable for assessing the outcomes of utilizing AI-based chatbots in digital health. As noted, cross-disciplinary and cross-sectoral collaboration is essential for addressing domain-specific AI applications and challenges, including ethical concerns [48]. This highlights the need for researchers and practitioners in medical science to collaborate with experts in communication and computer science [29].

Our findings partly align with a systematic review by Bedi et al. [49], as both highlight the need to train AI language models for specific clinical domains, particularly using real patient care data to ensure alignment with clinical conditions. They also call for greater generalizability in evaluating the effectiveness of AI-based chatbots, the development of standardized task definitions, and strategies for mitigating biases. In addition, our review highlighted the lack of randomised controlled trials (RCTs) and the limited use of theoretical frameworks in evaluating user engagement. A validated method is also needed for comparing different AI models across studies. Training AI language models for specific clinical domains (e.g., mental health consultation, clinical diagnosis) is crucial for health communication researchers. Achieving this requires future technological developments to better understand human emotions and generate more personalised outputs. These findings provide insight into the current state of English-language research and suggest future research avenues for AI-based chatbots in health.

We also observed that chatbot safety, while recognized as a high-priority area, is underrepresented—appearing in only 5% of the reviewed articles—highlighting a significant research gap that warrants urgent investigation. The review further indicates that these safety concerns primarily stem from AI models' limited understanding of human emotion and their lack of timely responses to potential risks [35].Further approach might be needed to address a broader understanding of safety, such as data security, privacy, and open data sharing in healthcare, one of the significant ethical issues raised in this systematic review, which call for advanced technical solutions [50]. In addition, a care ethics approach, which advocates for fostering a caring environment that takes account of context and circumstances  [Weinberger, 2024] , including the application of AI technology within specific political, economic, organizational, and personal contexts, is important for research in this area [48]. This approach enables the contextualisation of technological ethics to address ethical and safety concerns within specific domains [48], which helps promote the responsible use of AI-based chatbots in digital health.

The limitations of this review include the relatively small number of studies on AI-based chatbots. We also acknowledge that emerging AI models may restrict or protect patient information entered into algorithms, creating barriers for external researchers. Additional limitations include study selection criteria—for example, studies lacking MeSH "health" terms may

PLOS Digital Health

have been missed in PubMed—the short time frame (2022–2024), and the rapid evolution of AI models, whose capabilities have improved substantially since their initial release. It is also notable that the tools captured in the literature are not the only AI-based chatbots available for healthcare purposes. Many publicly accessible, health-focused GPTs exist outside the peer-reviewed literature, highlighting that numerous such chatbots operate beyond the scope of published studies [51]. While these fall outside the inclusion criteria of this systematic review, we acknowledge their presence as contextual evidence of the broader and rapidly evolving chatbot ecosystem. Future research, including studies in non-English contexts, could address these gaps by conducting more generalizable investigations of chatbot effectiveness, developing domain-specific AI models, and examining ethical and regulatory considerations for AI in healthcare.

## Supporting information

**S1 Appendix. PRISMA 2020 Checklist.** From: [15]. This work is licensed under CC BY 4.0. To view a copy of this license, visit https://creativecommons.org/licenses/by/4.0/.
(DOCX)

**S2 Appendix. The list of 348 articles from screening.**
(DOCX)

## Author contributions

**Conceptualization:** Shi Feng, Xiufang (Leah) Li.

**Data curation:** Shi Feng, Xiufang (Leah) Li.

**Formal analysis:** Shi Feng.

**Investigation:** Alexandra Nicole Wake.

**Methodology:** Shi Feng, Xiufang (Leah) Li.

**Project administration:** Alexandra Nicole Wake.

**Resources:** Alexandra Nicole Wake.

**Software:** Shi Feng.

**Supervision:** Xiufang (Leah) Li, Alexandra Nicole Wake.

**Validation:** Xiufang (Leah) Li.

**Writing – original draft:** Shi Feng.

**Writing – review & editing:** Xiufang (Leah) Li, Alexandra Nicole Wake.

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
