## [Decision Letter · Decision Letter 0]

6 Nov 2025

Response to Reviewers
Revised Manuscript with Track Changes
Manuscript
**Journal Requirements:**
**Additional Editor Comments (if provided):**
**Reviewers' Comments:**

**Comments to the Author**

1. Does this manuscript meet PLOS Digital Health’s publication criteria?

Reviewer #1: Yes

Reviewer #2: Yes

Reviewer #3: Partly

2. Has the statistical analysis been performed appropriately and rigorously?

Reviewer #1: Yes

Reviewer #2: I don't know

Reviewer #3: Yes

3. Have the authors made all data underlying the findings in their manuscript fully available (please refer to the Data Availability Statement at the start of the manuscript PDF file)?

Reviewer #1: Yes

Reviewer #2: Yes

Reviewer #3: No

4. Is the manuscript presented in an intelligible fashion and written in standard English?

Reviewer #1: Yes

Reviewer #2: Yes

Reviewer #3: Yes

Reviewer #1: 1. The abstract should more clearly state the number of articles contained in the total number collected (i.e., "A total of 348 articles were collected, 20 of which were analyzed.") to facilitate immediate comprehension.

2. The article refers to the limited theoretical framework discussion but fails to detail well enough why that constitutes a limitation or make concrete suggestions regarding which frameworks (beyond the couple of them they did mention) would be most beneficial for future engagement and efficacy studies to utilize.

3. While the paper categorizes AI models (Table 4) and mentions LLMs like GPT-3.5 and GPT-4, the separation of "AI-based chatbots specifically for healthcare" (Group 1) from "AI models of ChatGPT" (Group 2) and "Compared different AI models" (Group 3) could be clearer, especially when discussing technical foundations (e.g., the exact position of LLMs over trained data libraries in Group 1).

4. The article correctly lists chatbot safety as a topmost area, but also suggests that it only covers 5% of the articles covered, and there's a vast research gap around the same. This underrepresentation warrants more emphasis in the Discussion as an area of high priority, urgent investigation. The issue of a lack of wider applicability and generalisability (RQ4) is a reiterative topic but seems to be slightly repetitive across the limitations and future directions.

6. The usage of terms such as "AI-based chatbots" and "conversational agents" should be standardized. As defined, sometimes the difference is not clear-cut. For example, in the Introduction, it would be helpful to clearly declare that the emphasis will be on the AI-based subset once the two categories are introduced.

7. It directly addresses issues of data security, privacy, and open data sharing in healthcare, one of the significant ethical issues raised in the systematic review. It presents an advanced technical solution (Blockchain + MOA) for issues present in current AI models. It is suggested to refer from the paper: “Blockchain-Enabled Healthcare Optimization: Enhancing Security and Decision-Making Using the Mother Optimization Algorithm”.

8. Integrate Recent AI Technical Developments as the article mentions potential future research directions, such as the creation of LLMs designed for a particular clinical area. This section (RQ4) would be greatly enhanced by including technical concepts behind such advancements, e.g., utilizing specialized AI algorithms or blockchain in secure data processing, which is relevant directly to the "safety" and "data security" concerns identified. Action: Include a reference to an AI or machine learning approach which addresses the data security, efficiency, or computational complexity problems of AI in healthcare.

Reviewer #2: As data was not measured statistically, there are no statistical methods to address. Remainder of comments and review are included in attached document.

Reviewer Response:

This study provides a systematic review and descriptive summary of studies in healthcare that have assessed the role for AI-based chatbots since the emergence of ChatGPT in 2022, inclusive of September 2024. The authors carried out this study with validated PRISMA guidelines and registered it in the PROSPERO database.

The researchers state their goals for this study with 4 research questions (“RQs”). These research questions were designed to provide a broad descriptive overview of AI-chat bots in the field of health, the methods used in the available studies, key concerns, and noted future directions.

Due to heterogeneity of studies included, the authors are unable to perform a meta-analysis however do provide relative rates of aspects of studies such as research study type, target audience. The majority of the results are descriptive and include anecdotes from the various study author conclusions.

As this is an emerging technology, having a summary of various uses of the technology, along with success and failure can help direct future research. Some of the claims made by the authors in the introduction, and in the results/discussion are not supported by the stated references. These concerns as well as other thoughts/suggestions are listed here:

Introduction paragraph 1:

I am uncertain about why COVID-19 would accelerate the adoption of chatbots. With the development of large language models, there have been an increase in power for chat bots, although this development is independent of the virus. I could not find any data in Bombard et al, 2022 to support this claim.

Likewise, I am uncertain if the claim ‘widely used in healthcare’ can be justified. Are these widely used by providers or by patients? Are there any references to support the growth of use in this sector?

Table 2. Citation #7 and citation #20 are focusing on residency applications and anatomy education. Although reasonable to include these, the studies appear more geared towards academic education and administration rather than directly on digital health. It may be helpful to note in your methods that educational content was not excluded. Of course, the one limitation would be that there may be a number of chatbot-based medical school citations that were not included in your query and potential exclusion of education studies that don’t have MeSH “health” terms may have been missed.

Page 13: Paragraph 3. The authors state that their findings demonstrate a growing demand for transitioning from rule-based to AI-based chat bots. Their methods did not assess this claim, as they excluded any study using rule-based chat bots so it is unclear if demand is growing or receding. This paragraph could be better stated as just describing the two themes that the authors proposed are shown in the available studies (i.e. omit the final sentence and remove terms such as trends or growing demand).

Page 14: Paragraph 3. The authors state that “patients and everyday individual have adopted AI-based chatbots for various purposes”. The research studies cited do not all describe patient adaptation. For example, Lim et al (2024) had investigators enter prompts into commercially available large language models and authors analyzed the output, so this was not a study of patient adaptation. In addition Gordon et al. (2024) did a similar study where fictitious applications were developed by investigator and admissions faculty were surveyed. This did not show that this was being adopted into clinical practice. Patient adopting the technology would need to be demonstrated by patients using a technology spontaneously or sharing interest in a survey-based study, measured by looking at patient use, satisfaction, plan for future use etc.

Page 19: Paragraph 1. One of the major cited concerns with using large language model AI chat bots is the risk of “hallucination” (fabrication of data or answers to address the prompt when an obvious answer cannot be cited). In the final you state that most of the quantified studies found the text quality accurate and evidence-based. Are you able to summarize the quantified findings on the degree of accuracy. Are you able to quantify how often these studies noted hallucinations? If absolute numbers are hard to elucidate, at least some statement on the studies’ noting of hallucinations should be addressed.

Page 22: Paragraph 2. The authors state that “larger randomized controlled trials in more diverse cultural and demographic settings are needed”. To address which questions in particular? As their systematic review included studies from the earliest incorporation Chat GPT (page 5), one would expect most of the studies to be small at this time. Assessment of safety with smaller studies is appropriate for an emerging technology. Although I do not disagree that larger studies with broader demographics would increase our understanding, I would suggest rephrasing this statement to acknowledge the nascent nature of the technology and priority on ensuring accuracy and safety.

Due to the very recent emergence of Chat GPT, and the number of restrictions with protected patient information being entered into an algorithm that by nature learns and retains all inputs, initial studies using commercial large language models would not contain protected patient health information. This should be acknowledged by the authors as well as whether or not there are emerging large language models that protect patient privacy that overcome this barrier to research. Additional limitations beyond just the small number of studies should be acknowledged by the authors, including limitations of study selection, short time frame (2022-2024), as well as rapid development of the Chat GPT models (capabilities of these tools are vastly improved today compared to their initial introduction).

I would like to commend the authors for the undertaking and attempt to find a signal among noise in the emerging studies with AI chat bots. I would recommend ensuring citations support statements in the body of the text and clarity on whether education/administration was a focus on their study in addition to addressing the other queries above.

Reviewer #3: This manuscript represents a thoughtful attempt to conduct a systematic review on a complex and timely topic: AI-based chatbots in digital health. The authors follow PRISMA guidelines and apply a structured approach to identify and analyse studies evaluating AI-based chatbots. Comments below:

A key contextual limitation is that the work appears to have been undertaken around the same time as the systematic review by Bedi et al., published in JAMA (“Testing and Evaluation of Healthcare Applications of Large Language Models”). While the JAMA review is broader in scope, it employs a clearly rationalised search strategy and presents a framework for classifying and synthesising LLM research that is highly relevant here. The present paper would benefit from explicitly acknowledging and referencing that work, and ideally aligning aspects of its analysis and synthesis with it. In the present manuscript, Feng and colleagues identify four critical domains being evaluated by AI researchers, whereas Bedi et al. highlight several additional important concepts. The authors need to clarify how their review provides distinct or deeper insights within the narrower domain of AI-based health chatbots. At times, the paper appears less detailed in areas where greater specificity could add value – this is not to say in other areas they don’t provide more detail.

Some inconsistency in the terminology and framing of “AI-based chatbots” warrants attention throughout—particularly the implicit suggestion that the tools captured in the literature represent the only available health chatbots. For instance, our recent Frontiers in Public Health (https://www.frontiersin.org/journals/public-health/articles/10.3389/fpubh.2025.1584348/full) paper identified more than a thousand publicly accessible, health-focused GPTs, highlighting that many such chatbots exist outside the peer-reviewed literature. While these may fall beyond the inclusion criteria of this systematic review, their existence should be acknowledged as contextual evidence of the broader, evolving chatbot ecosystem.

The supplementary material could be improved to better share the search process and ensure that all data underlying the findings in their manuscript are fully accessible. The English is generally clear, but there are some difficulties in following the specified aims and layout of the results (and methods), which affects readability from my perspective.

Finally, although the paper identifies four main research areas—text quality, clinical efficacy, user engagement, and safety—the treatment of “safety” appears largely focused on response accuracy, with limited consideration of the broader safety implications of chatbot use. This is chiefly because such papers, such as the above two, were not part of the literature captured by the review conducted herein.

**Do you want your identity to be public for this peer review?** For information about this choice, including consent withdrawal, please see our Privacy Policy

Reviewer #1: No

Reviewer #2: No

Reviewer #3: **Yes:** Ashley M Hopkins

**Figure resubmission:**

**Reproducibility:** To enhance the reproducibility of your results, we recommend that authors of applicable studies deposit laboratory protocols in protocols.io, where a protocol can be assigned its own identifier (DOI) such that it can be cited independently in the future. Additionally, PLOS ONE offers an option to publish peer-reviewed clinical study protocols. Read more information on sharing protocols at https://plos.org/protocols?utm_medium=editorial-email&utm_source=authorletters&utm_campaign=protocols

---

## [Decision Letter · Decision Letter 1]

5 Jan 2026

Engaging Artificial Intelligence (AI)-Based Chatbots in Digital Health: A Systematic Review

PDIG-D-25-00805R1

Dear Dr Li,

We are pleased to inform you that your manuscript 'Engaging Artificial Intelligence (AI)-Based Chatbots in Digital Health: A Systematic Review' has been provisionally accepted for publication in PLOS Digital Health.

Best regards,

Danielle Bitterman

Guest Editor

PLOS Digital Health

**Additional Editor Comments (if provided):**

**Reviewer Comments (if any, and for reference):**

Reviewer's Responses to Questions

**Comments to the Author**

Reviewer #1: All comments have been addressed

Reviewer #2: All comments have been addressed

Reviewer #3: All comments have been addressed

publication criteria?

Reviewer #1: Yes

Reviewer #2: Yes

Reviewer #3: Yes

3. Has the statistical analysis been performed appropriately and rigorously?

Reviewer #1: Yes

Reviewer #2: N/A

Reviewer #3: Yes

4. Have the authors made all data underlying the findings in their manuscript fully available (please refer to the Data Availability Statement at the start of the manuscript PDF file)?

Reviewer #1: Yes

Reviewer #2: Yes

Reviewer #3: Yes

5. Is the manuscript presented in an intelligible fashion and written in standard English?

Reviewer #1: Yes

Reviewer #2: Yes

Reviewer #3: Yes

Reviewer #1: The revised manuscript is a thorough and well-executed systematic review that fits the journal's focus. The authors have addressed all earlier concerns and added important details that significantly enhance the paper's quality. I believe the manuscript is ready for acceptance.

1. The authors have provided clear, point-by-point responses that show they have incorporated all requested changes from the previous review. These revisions include important improvements across the Abstract, Introduction, Methods, and Discussion sections, showing their responsiveness to feedback.

2. The manuscript includes a more detailed discussion about the lack of theoretical engagement in previous research. Additionally, the authors have incorporated theoretical concepts, such as Uses and Gratifications Theory and Care Ethics, to propose robust frameworks for future user engagement and ethical research.

3. The authors have resolved earlier confusion about classifying and understanding different AI models. Table 5 now clearly explains how healthcare-specific AI-based chatbots combine generative parts with trained data libraries (e.g., Wysa, QuitBot), establishing a clear technical context for their study sample.

4. The Discussion section has been greatly expanded to highlight the lack of research on chatbot safety as an urgent area for investigation. The authors discuss the need for a broader understanding of safety, including data security and privacy concerns, and call for advanced technical solutions, such as blockchain integration.

5. The authors justified the inclusion of medical education and administration in the field of digital health in the Methods section, addressing concerns about study selection. They also included a summary of important findings regarding text quality and hallucination risks, which improves the rigor of the results section.

Reviewer #2: All suggestions addressed.

Reviewer #3: The authors have responded constructively to the reviewers, with improvements to the Introduction and Discussion that better situate the work in the existing literature and more clearly acknowledge key constraints (including the dependence of findings on the search strategy, MeSH terms, and timeframe). The revision is overall clearer.

I would suggest a small further revision to strengthen alignment between the paper’s conclusions (the four research areas for AI chatbots: text quality, clinical efficacy, user engagement, and safety) and the wider evaluation domains outlined in the JAMA framework paper (“Testing and Evaluation of Healthcare Applications of Large Language Models”). Specifically, the authors could more explicitly summarise that Bedi et al highlight the need for research addressing accuracy, comprehensiveness, factuality, robustness, fairness, bias, toxicity, calibration, uncertainty, and deployment metrics such as cost, latency, and runtime, and briefly note how these domains complement and extend the review’s four areas. This is a minor but warranted clarification for framing (notwithstanding the authors’ clear efforts to incorporate this literature already).

**Do you want your identity to be public for this peer review?** For information about this choice, including consent withdrawal, please see our Privacy Policy

Reviewer #1: No

Reviewer #2: No

Reviewer #3: **Yes:** Ashley Hopkins
